# MoMu-Diffusion: On Learning Long-Term Motion-Music Synchronization and Correspondence

**Fuming You, Minghui Fang, Li Tang, Rongjie Huang, Yongqi Wang, Zhou Zhao**[*]
Zhejiang University
fumyou13@gmail.com

## Abstract

Motion-to-music and music-to-motion have been studied separately, each attracting substantial research interest within their respective domains. The interaction between human motion and music is a reflection of advanced human intelligence, and establishing a unified relationship between them is particularly important. However, to date, there has been no work that considers them jointly to explore the modality alignment within. To bridge this gap, we propose a novel framework, termed MoMu-Diffusion, for long-term and synchronous motion-music generation. Firstly, to mitigate the huge computational costs raised by long sequences, we propose a novel Bidirectional Contrastive Rhythmic Variational Auto-Encoder (BiCoR-VAE) that extracts the modality-aligned latent representations for both motion and music inputs. Subsequently, leveraging the aligned latent spaces, we introduce a multi-modal Transformer-based diffusion model and a cross-guidance sampling strategy to enable various generation tasks, including cross-modal, multi-modal, and variable-length generation. Extensive experiments demonstrate that MoMu-Diffusion surpasses recent state-of-the-art methods both qualitatively and quantitatively, and can synthesize realistic, diverse, long-term, and beat-matched music or motion sequences. The generated samples and codes are available at https://momu-diffusion.github.io/.

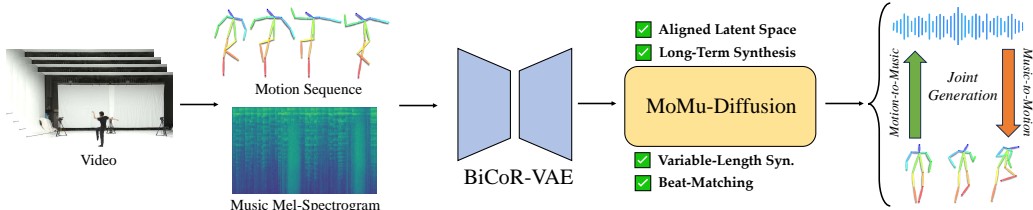

Figure 1: The pipeline of MoMu-Diffusion. MoMu-Diffusion integrates the alignment of motion and music through the novel Bidirectional Contrastive Rhythmic Auto-Encoder (BiCoR-VAE). Leveraging the aligned latent space, MoMu-Diffusion facilitates both cross-modal and multi-modal generations.

## 1 Introduction

Dancing to the musical beats or creating a variety of rhythmically synchronized music for a given motion is a fundamental aspect of human creativity. Music and human motions serve as universal languages that are shared by all civilizations, transcending cultural and geographical boundaries around the world [25]. For computational methodologies, the motion-music generation poses several challenges: 1) maintaining long-term coherence in typically lengthy motion-music sequences 2)

---

[*]Corresponding Author

38th Conference on Neural Information Processing Systems (NeurIPS 2024).

Table 1: Comparison with the state-of-the-art audio-visual generation works, including but not limited to motion-music generation.

| Method | Pub. | Joint Generation | Pretrain | Long-Term Synthesis | Latent Space |
|--------|------|:----------------:|:--------:|:-------------------:|:------------:|
| Diff-Foley | NeurIPS'23 | ✗ | ✓ | ✗ | ✓ |
| MM-Diffusion | CVPR'23 | ✓ | ✗ | ✗ | ✗ |
| LORIS | ICML'23 | ✗ | ✗ | ✓ | ✗ |
| D2M | NeurIPS'19 | ✗ | ✓ | ✗ | ✓ |
| CDCD | ICLR'23 | ✗ | ✗ | ✓ | ✓ |
| MoMu-Diffusion | | ✓ | ✓ | ✓ | ✓ |

ensuring temporal synchronization and rhythmic alignment between motion and music sequences, and 3) generating realistic, diverse, and variable-length human motions or music.

Existing works usually divide the motion-music generation into two distinct tasks: motion-to-music and music-to-motion. For motion-to-music, some methods compress the conditional video frames into a single image, in which the temporal information is lost [52, 53]. The state-of-the-art work, LORIS [49], employs a hierarchical conditional diffusion model to generate long-term musical waveforms. However, LORIS introduces huge computational costs and training difficulties since it generates long-term musical waveforms directly. For music-to-motion, the Dancing2Music (D2M) [26] framework divides the generation process into two stages: decomposing the dance into basic dancing movements with a VAE and compositing the basic movements into dance with a GAN. Nonetheless, D2M's approach of segmenting long-term music into short clips (approximately 1-2 seconds) diminishes the coherence of the synthesized motion sequences.

Motivated by the fact that human motions are highly associated with music yet existing computational methods often study them in isolation, we propose a novel multi-modal framework, termed MoMu-Diffusion, to address the aforementioned challenges jointly. Firstly, to mitigate the computational costs and optimization complexities raised by long sequences, we employ a VAE to encode both motion and music sequences into latent spaces. Subsequently, to investigate the relationship between human movements and musical beats, we propose rhythmic contrastive learning. This approach involves constructing contrast pairs with a kinematic amplitude indicator, which quantifies the temporal variation in motion and is derived from the spatial motion directrogram differences as detailed in [4]. Given that the motion and music sequences are interactively aligned in the latent space to discern the correlation between kinematic shifts and musical rhythmic beats, we call our model as the Bidirectional Contrastive Rhythmic VAE (BiCoR-VAE).

With the aligned latent space, we introduce a Transformer-based diffusion model that captures long-term dependencies and facilitates sequence generation across variable lengths. Additionally, we introduce a simple cross-guidance sampling strategy that integrates different cross-modal generation models, enabling multi-modal joint generation without extra training. By incorporating the BiCoR-VAE and the diffusion Transformer model, our MoMu-Diffusion framework effectively models the long-term motion-music synchronization and correspondence, enabling motion-to-music, music-to-motion, and joint motion-music generation. Moreover, MoMu-Diffusion supports generating motion-music samples in variable lengths. The pipeline of MoMu-Diffusion is illustrated in Figure 1.

We have conducted extensive experiments on three motion-to-music and two music-to-motion datasets, including scenarios such as dancing and competitive sports. The experimental results demonstrate that MoMu-Diffusion attains state-of-the-art performance across both objective and subjective metrics, significantly enhancing music/motion quality and cross-modal rhythmic/kinematic alignment. Furthermore, we have carried out abundant ablation studies to validate the efficacy of the BiCoR-VAE and the DiT architecture. A comparative analysis with state-of-the-art motion-to-music methods CDCD [53] and LORIS [49], 2D music-to-motion method D2M [26], and general video-to-audio methods Diff-Foley [33] and MM-Diffusion [41], is presented in Table 1.

## 2   Related Works

**Neural Motion Synthesis.** Neural motion synthesis is often associated with audio, and we focus on two audio-driven scenarios: music-to-motion generation [12, 27, 35, 26]and co-speech gesture generation [48, 32, 50]. For music-to-motion, some methods [12, 27, 35] propose to retrieve the most related music for the given motion sequence. D2M [26] is a generative model that designs

a "decomposition-to-composition" method to learn the movement units and generate music from the learned units. Besides, some methods [30, 54, 29, 1] investigate synthesizing 3D motions from music. For co-speech gesture generation, DiffGesture [50] is a state-of-the-art model with a diffusion transformer architecture and diffusion gesture stabilizer. We study the 2D music-to-motion problem and compare the proposed MoMu-Diffusion with DiffGesture and D2M

**Neural Music Synthesis.** Neural music Synthesis aims to generate melodious music with generative neural networks. Various generative models have been successfully applied to music synthesis such as transformer-based autoregressive models [21, 39, 9], VAE [2, 40, 6], GAN [8, 24, 36], and diffusion models [16, 34]. Some efforts have been made to video-to-music which focuses on the cross-modal temporal alignment. For example, Foley Music [13] and Audeo [43] utilize Musical Instrument Digital Interface (MIDI) representations to generate music in a non-regressive manner. D2M-GAN [52] and CDCD [53] generate video-related music by compressing the video frames into a single image, in which the temporal information is neglected. LORIS [49] proposes a hierarchical conditional diffusion model to generate long-term musical waveforms.

**Multi-Modal Contrastive Learning.** Contrastive has been demonstrated effective in For example, Elizalde et al. [10] proposed Contrastive Language-Audio Pretraining (CLAP) to learn a unified latent representation for an audio or text input, facilitating the birth of text-to-audio models [31, 20]. For audio-visual generative tasks, DiffFoley [33] uses semantic and temporal contrastive learning to promote video-to-audio generation. In this paper, to improve the efficiency and generalization ability of our generative model, we propose the first motion-music pretraining model with a well-designed contrastive loss to learn beat synchronization and rhythm correspondence.

## 3 Bidirectional Contrastive Rhythmic VAE (BiCoR-VAE)

### 3.1 Multi-Modality Model Architecture

**Motion Variational Auto-Encoder.** Let $n \in \mathbb{R}^{T_m \times J \times 2}$ be the 2D motion keypoints extracted from the corresponding video, where $T_m$ is the motion frames, $J$ is the number of nodes containing the values of the $x$-coordinate and $y$-coordinate. Then, we encode the spatial positions into a latent by $z_m = E_m(m) \in \mathbb{R}^{T_{zm} \times d}$, where $T_{zm} < T_m$ is the downsampled motion frames and $d$ is the latent motion dimension. The encoded latent can be decoded by a decoder to obtain the reconstructed motion sequence: $m' = D_m(m)$.

**Music Variational AutoEncoder.** Music is a structured and complex audio signal, composed of various elements such as melody, harmony, rhythm, and dynamics. Some works [13, 43] utilize Musical Instrument Digital Interface (MIDI) representations, which yield highly formulated results. However, processing long-term music directly from the raw waveform is computationally intensive and challenging [49]. To address this, we train a VAE on the mel-spectrogram derived from the music, coupled with a high-fidelity vocoder. Let $u \in \mathbb{R}^{T_u}$ be a music input, where $T_u$ denotes the waveform length. We can extract the mel-spectrogram of the music input: $a = Mel(u) \in \mathbb{R}^{C_a \times T_a}$, where $Mel()$ is the pre-defined mel-spectrogram extraction function, $C_a$ is the channels, and $T_a \ll T_u$ is the frames. Then, an encoder is used to compress the mel-spectrogram into a latent: $z_a = E_a(a) \in \mathbb{R}^{T_{za} \times d}$, where $T_{za}$ is the downsampled music frames and $d$ is the latent mel-spectrogram dimension. The encoded mel-spectrogram can be decoded by a decoder $a' = D_a(a)$, and subsequently the musical waveform can be obtained by a high-fidelity vocoder $x' = V(a')$.

### 3.2 Rhythmic Contrastive Learning

Contrastive learning has proven effective for learning multi-modal representations, enhancing performance in downstream tasks [38]. In the context of temporal alignment, a recent work [33] introduces temporal contrast, which seeks to maximize the similarity of audio-visual pairs from the same time segment while minimizing the similarity of pairs from different segments. However, this paradigm faces limitations in long-term motion-music synthesis, as musical pieces typically correspond to numerous rhythmic beats. The random selection process for constructing negative samples risks capturing similar rhythmic sequences, which undermines the learning objective. To address it, we propose rhythmic contrastive learning, designed to align cross-modal temporal synchronization and rhythmic correspondence. Based on the motion and music VAEs, we can obtain the motion latent $z_m \in \mathbb{R}^{T_{zm} \times d}$, and music mel-spectrogram latent $z_a \in \mathbb{R}^{T_{za} \times d}$, respectively. To synchronize the

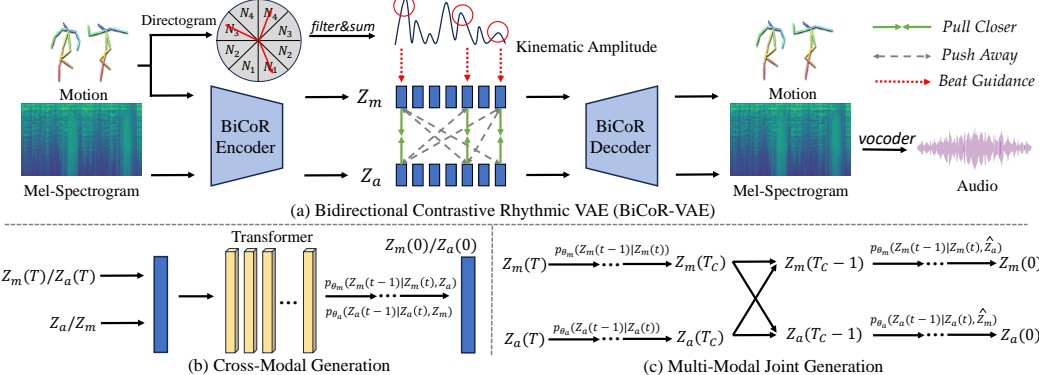

Figure 2: An overview of the proposed MoMu-Diffusion framework. MoMu-Diffusion contains two integral components: a bidirectional contrastive rhythmic Variational Autoencoder (BiCoR-VAE) designed to learn the aligned latent space, and a Transformer-based diffusion model responsible for sequence generation. This framework is adept at facilitating both cross-modal and multi-modal joint generations, offering a robust approach to the integrated synthesis of motion and music.

motion and music, which are often sampled differently, we employ pre-processing techniques such as evenly dropping motion frames to match the number of music frames, ensuring that $T_{zm} = T_{za}$.

In the domain of motion-guided music, the inherent irregularity of human movements, characterized by rapid and abrupt actions, can significantly influence rhythm. To synchronize these rhythmic patterns, we employ a kinematic amplitude indicator as a basis for constructing contrastive clips within each motion-music pair. Firstly, we extract the motion kinematic offsets [15] with the motion directogram [4], a metric that quantifies the variation in motion. We denote $F(r, j)$ as the first-order difference of $j$-th node in the 2D motion at temporal timestep $r$, and divide it into $K$ bins based on their Euclidean angles with $x$-axis by $\tan^{-1}(y/x)$. Then, the 2D motion directogram $D(r, \theta)$ can be expressed as the aggregate of $F(r, j)$ across each angular bin:

$$D(r, \theta) = \sum_{j=1}^{J} ||F(r, j)||_2 \mathbb{1}_\theta(\angle F(r, j)), \quad \text{where } \mathbb{1}_\theta(\phi) := \begin{cases} 1, & |\theta - \phi| \leq 2\pi/K, \\ 0, & \text{otherwise.} \end{cases} \quad (1)$$

The indicator function $\mathbb{1}_\theta(\phi)$ distributes the motion nodes into $K$ angular bins. Then, the kinematic amplitude indicator is computed by summing the bin-wise directogram difference in each angular column:

$$Q(r) = \sum_{k=1}^{K} \max(0, |D(r, k)| - |D(r - 1, k)|), \quad (2)$$

where $D(r, k)$ is the directogram volume at temporal timestep $r$ and $k$-th bin. The kinematic amplitude value is normalized within the range of (0,1).

With the kinematic amplitude indicator established, we proceed to prepare the temporal motion-music clips for contrastive rhythmic learning. For each motion-music latent pair, we randomly sample $N_T$ motion-music clips and divide them into $N_C$ categories according to the clip-wise maximum kinematic amplitude values. In order to maximize the similarity of motion-music pairs from the same timestep (i.e. temporal alignment) and minimize the similarity of motion-music pairs across different timesteps and rhythmic patterns, we randomly sample $N_S$ motion-music latent clip $(c_a^{r_s:r_e}, c_m^{r_s:r_e}, Q(r_s:r_e)) \in (\mathbb{R}^d, \mathbb{R}^d, (0, 1))$ from different kinematic amplitude categories for the temporal and rhythmic alignment:

$$c_a^{r_s:r_e} = P_{\max}(z_a^{r_s}:z_a^{r_e}), \ c_m^{r_s:r_e} = P_{\max}(z_m^{r_s}:z_m^{r_e}), \ Q(r_s:r_e) = \max(Q(r_s):Q(r_e)), \quad (3)$$

where $r_s$ and $r_e$ denote the start and end timesteps of the sampled clip, respectively, and $P_{\max}$ denotes the max-pooling operation across the temporal dimension. Finally, based on the sampled motion-music clips $\{(c_a^i, c_m^i)\}_{i=1}^{N_C}$, the contrastive objective can be formulated as:

$$\mathcal{L}_{\text{contrast}} = -\frac{1}{2} \log \frac{\exp(sim(c_a^i, c_m^j)/\tau)}{\sum_{c=1}^{N_C} \exp(sim(c_a^i, c_m^c)/\tau)} - \frac{1}{2} \log \frac{\exp(sim(c_a^i, c_m^j)/\tau)}{\sum_{c=1}^{N_C} \exp(sim(c_a^c, c_m^j)/\tau)}. \quad (4)$$

## 3.3 Training Strategy

In BiCoR-VAE, the goal is to learn two paired VAEs for motion and music inputs, with a focus on temporal and rhythmic alignment within the low-level latent space. However, the VAE's objective to preserve fine-grained details for accurate reconstruction often conflicts with contrastive rhythmic learning's aim to align latent representations across modalities. This presents a trade-off between representational fidelity and generative alignment, posing optimization challenges. To address it, we propose a two-stage training strategy: initially, we train the music VAE using both a VAE loss and a GAN loss to prevent over-smoothing of the mel-spectrogram; subsequently, we train the motion VAE with a VAE loss and the contrastive rhythmic loss, while keeping the music VAE's parameters fixed. The insight behind this strategy is that mel-spectrograms, with their rich and complex acoustic features, require a more intricate optimization process compared to motion VAE, which deals with a limited set of body joint data. An overview of BiCoR-VAE is illustrated in Figure 2 (a).

## 4 Transformer-based Diffusion Model with Aligned BiCoR-VAE

**Diffusion Formulation.** Recent works have revealed that the U-Net architecture is not essential for diffusion probabilistic modeling, and in fact, the transformer can achieve superior performance in text-to-image generation tasks [37, 11]. Additionally, the transformer architecture excels at capturing long-range dependencies within sequence data and offers flexibility for variable-length generation [45]. Inspired by these findings, we opt for a Transformer-based architecture for our motion-music generation framework. Concretely, our approach involves initially concatenating the noisy input with the embedded conditional inputs and the embedded diffusion timesteps along the temporal dimension. This fused input is then padded to match a specified maximum length and combined with positional embeddings prior to being processed by the DiT model. The DiT output is subsequently truncated to the original temporal length and mapped to the output latent space. To illustrate the diffusion process, let's consider the motion-to-music task. During the forward diffusion, the latent data is gradually perturbed towards a standard Gaussian distribution according to a pre-defined schedule $\alpha_1, ..., \alpha_T$, where $T$ is the total diffusion timesteps and $\overline{\alpha}_t = \prod_{i=1}^{t} \alpha_i$:

$$q(z_a(t)|z_a(t-1)) = \mathcal{N}(z_a(t); \sqrt{\alpha_t} z_a(t-1), 1 - \alpha_t \mathbf{I}), \tag{5}$$

where $z_a(t)$ denotes the music latent at timestep $t$. Then, the training objectives of our DiT-based cross-modal generation models are defined:

$$\mathcal{L}_{\text{m2a}} = ||\epsilon_{\theta_a}(z_a(t), t, z_m) - \epsilon||_2^2, \qquad \mathcal{L}_{\text{a2m}} = ||\epsilon_{\theta_m}(z_m(t), t, z_a) - \epsilon||_2^2, \tag{6}$$

where $\epsilon \in \mathcal{N}(0, 1)$ denotes the noise in diffusion procedure, $\theta_a$ and $\theta_m$ are the parameterized DiT denoisers for motion-to-music and music-to-motion generation, respectively.

**Conditional Generation.** For the cross-modal generation such as motion-to-music and music-to-motion, we implement classifier-free guidance [5, 18]. This method adeptly combines conditional and unconditional scores to obtain a trade-off between quality and diversity. By interpreting the diffusion model output as a score function, the sampling procedure with classifier-free guidance of motion-to-music can be written as:

$$\hat{\epsilon}_{\theta_a}(z_a(t), t, z_m) = \epsilon_{\theta_a}(z_a(t), t, \emptyset) + s \cdot (\epsilon_{\theta_a}(z_a(t), t, z_m) - \epsilon_{\theta_a}(z_a(t), t, \emptyset)) \tag{7}$$

where $s > 1$ denotes the classifier sampling scale to balance the diversity and quality of synthesized samples. The diffusion model with $\emptyset$ condition is achieved by randomly dropping $z_m$ and replacing it with an embedded "null" representation. Exchanging the latent inputs enables the sampling procedure for music-to-motion generation since we have built a modality-aligned latent space.

**Joint Generation with Cross Guidance.** To accomplish multi-modal joint generation, we propose a cross-guidance sampling strategy. This approach leverages multiple 'expert' models and introduces a slight modification to the sampling procedure, rather than integrating multiple modalities into a single model. Let $T$ be the total diffusion steps, $\epsilon_{\theta_a}$ be the trained motion-to-music denoising model, and $\epsilon_{\theta_m}$ be the trained music-to-motion denoising model, we perform unconditional generation before a defined diffusion step $T_c$:

$$p_{\theta_a}(z_a(t-1)|z_a(t)) = \mathcal{N}(z_a(t-1), \mu_{\theta_a}(z_a(t), t, \emptyset), \sigma_t^2 \mathbf{I}), \quad \text{where } T > t > T_c, \tag{8}$$

$$\mu_{\theta_a}(z_a(t), t, \emptyset) = \frac{1}{\sqrt{\alpha_t}}(z_a(t) - \frac{1 - \alpha_t}{\sqrt{1 - \overline{\alpha}_t}} \epsilon_{\theta_a}(z_a(t), t, \emptyset)), \quad \sigma^2 = \frac{1 - \overline{\alpha}_{t-1}}{1 - \overline{\alpha}_t}(1 - \alpha_t). \tag{9}$$

Table 2: Motion-to-music with **beat-matching** metrics.

| Subset | AIST++ Dance | | | | |
|---|---|---|---|---|---|
| Metrics | BCS↑ | CSD↓ | BHS↑ | HSD↓ | F1↑ |
| Foley | 96.4 | 6.9 | 41.0 | 15.0 | 57.5 |
| CMT | 97.1 | 6.4 | 46.2 | 18.6 | 62.6 |
| D2MGAN | 95.6 | 9.4 | 88.7 | 19.0 | 93.1 |
| CDCD | 96.5 | 9.1 | 89.3 | 18.1 | 92.7 |
| LORIS | **98.6** | 6.1 | 90.8 | 13.9 | 94.5 |
| Ours | 97.5 | **5.2** | **98.6** | **2.8** | **98.1** |

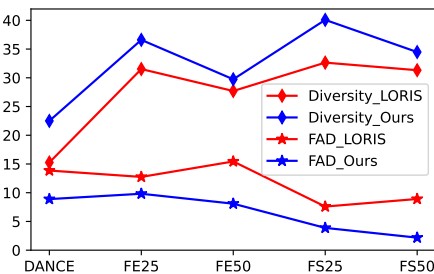

Figure 3: Motion-to-music with **generation quality** metrics: FAD↓ and Diversity↑.

| Subset | Floor Exercise-25s | | | | | Floor Exercise-50s | | | | |
|---|---|---|---|---|---|---|---|---|---|---|
| Metrics | BCS↑ | CSD↓ | BHS↑ | HSD↓ | F1↑ | BCS↑ | CSD↓ | BHS↑ | HSD↓ | F1↑ |
| Foley | 36.0 | 36.2 | 32.3 | 30.7 | 34.1 | 32.6 | 38.0 | 28.4 | 32.5 | 30.4 |
| CMT | 46.4 | 30.1 | 57.4 | 29.8 | 51.3 | 42.3 | 32.0 | 53.8 | 31.7 | 47.4 |
| D2MGAN | 45.3 | 27.7 | 58.7 | 30.1 | 51.1 | 41.9 | 29.2 | 54.7 | 32.7 | 47.5 |
| CDCD | 49.0 | 21.1 | 61.0 | 27.0 | 54.3 | 45.9 | 23.8 | 57.5 | 29.3 | 51.0 |
| LORIS | 58.8 | 19.4 | 67.1 | 21.1 | 62.7 | 54.7 | **21.6** | 63.8 | 24.5 | 58.9 |
| Ours | **66.6** | **14.3** | **76.9** | **19.1** | **71.4** | **62.7** | 24.0 | **68.1** | **20.2** | **65.3** |

Table 3: Results on the Floor Exercise dataset with **beat-matching** metrics.

Eq (8) and Eq (9) delineate the reverse process for motion-to-music generation within the timestep range $T \geq t > T_c$. The reverse process for music-to-motion generation can be similarly constructed. For reverse timesteps $T_c \geq t > 0$, we use the estimated clean motion/music latent to condition the generation process of music/motion with the classifier-free guidance defined in Eq (7). Given that the diffusion model adopts a coarse-to-fine refinement in the reverse process, we conduct unconditional generation before $T_c$ and impose conditional generation with the cross-guidance strategy after $T_c$, as the noise in the estimated clean latent is significantly reduced. Determining the value of $T_c$ appears to be quite challenging; however, our empirical findings indicate that the joint generation maintains robust performance across a broad range of values for $T_c$ (from $0.3T$ to $0.7T$). The diversity in joint generation is sustained by the unconditional process and classifier-free guidance. An overview of cross-modal generation and joint generation is shown in Figure 2 (b) and (c).

## 5 Experiments

### 5.1 Motion-to-Music Generation

**Experimental Settings.** We evaluate our method on the latest LORIS benchmark [49], which contains 86.43 hours of video samples synchronized with music. This benchmark presents three demanding scenarios: AIST++ Dance [30], Floor Exercise [42], and Figure Skating [47, 46]. In our experiments, each dataset is randomly split with a 90%/5%/5% proportion for training, validation, and testing. For model evaluation, we use five metrics to measure the beat-matching between synthesized music and ground-truth music [49]: Beats Coverage Scores (**BCS**) and Beat Hit Scores (**BHS**), Coverage Standard Deviation (**CSD**), Hit Standard Deviation (**CSD**), and the **F1** scores. Besides, we use the Fréchet Audio Distance (**FAD**) [22] and **Diversity** [26] scores to evaluate the quality of synthesized music. Since the quality of the Floor Exercise and the Figure Skating datasets are poor, we only conduct motion-to-music generation on them with a learnable motion encoder, whose architecture is derived from [51]. During sampling, we employ 50 DDIM sampling steps. More experimental settings are provided in Appendix B.

**Baselines.** We compare our proposed method to existing advanced video-to-music baselines: 1) **Foley Music** [13], a graph transformer framework with MIDI representations. 2) **CMT** [7], a controllable music transformer model to learn the rhythmic consistency between video and music. 3) **D2M-GAN** [52], a GAN-based model with vector quantized music representation. 4) **CDCD** [53], a diffusion-based model with an additional conditional discrete contrastive diffusion loss. 5) **LORIS** [49], a diffusion-based model with hierarchical conditional mechanism, yielding state-of-the-art performance on video-to-music synthesis.

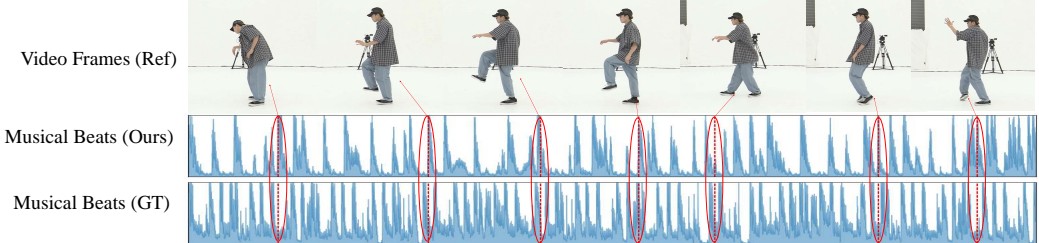

Figure 4: Example of beat matching on the motion-to-music generation. The red dashes indicate the extracted musical beats. The red arrow points to the video frame at that particular moment.

| Subset | Figure Skating-25s | | | | | Figure Skating-50s | | | | |
|---|---|---|---|---|---|---|---|---|---|---|
| Metrics | BCS↑ | CSD↓ | BHS↑ | HSD↓ | F1↑ | BCS↑ | CSD↓ | BHS↑ | HSD↓ | F1↑ |
| Foley | 36.0 | 36.2 | 32.3 | 30.7 | 34.1 | 32.6 | 38.0 | 28.4 | 32.5 | 30.4 |
| CMT | 46.4 | 30.1 | 57.4 | 29.8 | 51.3 | 42.3 | 32.0 | 53.8 | 31.7 | 47.4 |
| D2MGAN | 45.3 | 27.7 | 58.7 | 30.1 | 51.1 | 41.9 | 29.2 | 54.7 | 32.7 | 47.5 |
| CDCD | 49.0 | 21.1 | 61.0 | 27.0 | 54.3 | 45.9 | 23.8 | 57.5 | 29.3 | 51.0 |
| LORIS | 58.8 | 19.4 | 67.1 | **21.1** | 62.7 | 54.7 | **21.6** | 63.8 | 24.5 | 58.9 |
| Ours | **63.5** | **16.3** | **75.6** | 28.8 | **69.0** | **59.9** | 22.6 | **68.7** | **22.2** | **64.0** |

Table 4: Results on the Figure Skating with **beat-matching** metrics.

**Main Results.** The results of beat-matching are shown in Table 2, 3 and 4. From these tables, we can draw the following conclusions: 1) MoMu-Diffusion significantly surpasses existing state-of-the-art methods in cross-modal beat-matching. It demonstrates the effectiveness of BiCoR-VAE and the multi-modal Transformer-based model in synchronizing kinematic and rhythmic beats. 2) MoMu-Diffusion realizes a substantial improvement in Beat Hit Scores (BHS), which indicates the beats in the synthesized music are closely aligned with the ground truth. For example, MoMu-Diffusion gains 98.6% BHS on the AIST++ dancing subset, while previous methods usually gain about 90% BHS. An illustrative example of beat-matching for motion-to-music is presented in Figure 4. We can find the musical beats of synthesized music are aligned with the ground truth and the kinematic movements of the eference video.

The FAD and Diversity results are shown in Figure 3. In this comparison, we focus on LORIS, the current state-of-the-art method in motion-to-music generation. It is evident that MoMu-Diffusion consistently outperforms LORIS across these metrics, particularly in FAD scores. This superiority can be attributed to MoMu-Diffusion's architectural innovations for capturing long-term correspondence. Unlike text, music encompasses a richer sequence length due to its complex acoustic features, such as melody, rhythm, and driving beats. To address this, MoMu-Diffusion employs mel-spectrograms in place of raw waveforms, thereby mitigating sequence length. Additionally, the introduction of BiCoR-VAE facilitates modality alignment in latent spaces.

## 5.2 Music-to-Motion Generation

**Experimental Settings.** We use two datasets: AIST++ Dance [30] and BHS Dance. About 71 hours BHS Dance videos are collected from [26], which contains three dancing types: "Ballet", "Zumba", and "Hip-Hop". For model evaluation, we compute the beat-matching metrics between synthesized motion beats and the reference musical beats with the aforementioned five beat-matching metrics. To validate the quality of synthesized motion sequences, we use Fréchet Inception Distance (FID) [17], Mean KL-Divergence (Mean KLD), and the Diveristy scores. The feature extractor is based on MotionBert [51] and trained with a classification task on the BHS Dance dataset. For the BHS Dance dataset, we exclude the BiCoR-VAE since this dataset only contains the paired audio MFCC features and motion sequences without raw audio. In the generation process, the settings of the diffusion transformer model are the same as motion-to-music. More details are provided in Appendix B.

**Baselines.** We compare MoMu-Diffusion to two baselines: 1) **D2M** [26], the state-of-the-art music-to-motion work with a two-stage movement unit-based model; 2) **DiffGesture** [50], the state-of-the-art co-speech gesture generation work with a U-Net diffusion model. Dance Revolution [19] reports better performance on music-to-motion generation but is withdrawn by its authors.

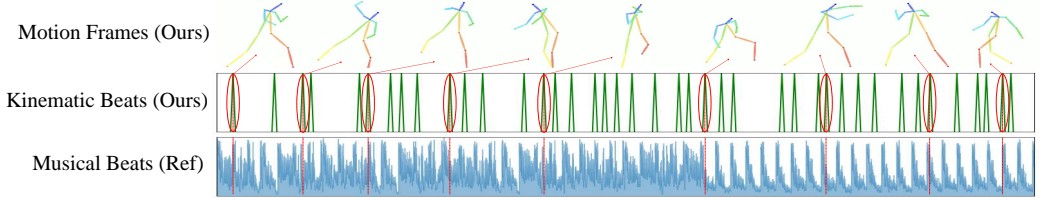

Figure 5: Example of beat matching on the music-to-motion generation. The red dashes indicate the extracted kinematic beats of the synthesized motion. The red arrow points to the frame of the synthesized motion sequence at that particular moment.

| Subset | AIST++ Dance | | | | | BHS Dance | | | | |
|--------|------|------|------|------|------|------|------|------|------|------|
| Metrics | BCS↑ | CSD↓ | BHS↑ | HSD↓ | F1↑ | BCS↑ | CSD↓ | BHS↑ | HSD↓ | F1↑ |
| D2M | 23.7 | 13.8 | 42.8 | 23.6 | 30.5 | 35.1 | 15.9 | 57.5 | 35.0 | 43.6 |
| DiffGesture | 28.5 | 16.7 | 40.4 | 25.7 | 33.4 | 42.8 | 21.3 | 61.1 | 23.9 | 50.3 |
| Ours | **39.2** | **10.2** | **56.3** | **12.0** | **46.2** | **47.9** | **8.4** | **78.5** | **12.1** | **59.5** |

Table 5: Results on the AIST++ Dance and BHS Dance datasets with **beat-matching** metrics.

**Main Results.** The beat-matching results are detailed in Table 5. An analysis of these results reveals that MoMu-Diffusion achieves superior scores across all evaluated tasks, outperforming the state-of-the-art music-to-motion method D2M and co-speech gesture generation method DiffGesture. This performance underscores the efficacy of our BiCoR-VAE in constructing an aligned latent space for cross-modal generation and the feed-forward diffusion model in capturing long-term correspondence. It should be noted that the metrics BCS (Beats Coverage Scores) and BHS (Beat Hit Scores) are defined differently in this context compared to motion-to-music scenarios. Specifically, BCS calculates the coverage score between the kinematic beats of the synthesized motions and the musical beats of the ground-truth music, rather than the kinematic beats of the ground-truth motions.

The generation quality results are presented in Table 6. It is observable that MoMu-Diffusion reports better FID, Mean KLD, and Diversity scores on both the AIST++ and BHS Dance datasets. It demonstrates that MoMu-Diffsuion can generate more realistic and high-quality motion sequences while maintaining the capability of diverse generations. We further present a qualitative example of music-to-motion beat-matching in Figure 5. We can find the kinematic beats of synthesized motion are highly associated with the reference musical beats. Additionally, the generated dance exhibits a high degree of diversity, encompassing lateral movements, rotations, squats, and so on.

| Subset | AIST++ Dance | | | BHS Dance | | |
|--------|------|------|------|------|------|------|
| Metrics | FID↓ | Diversity↑ | Mean KLD↓ | FID↓ | Diversity↑ | Mean KLD↓ |
| D2M | 17.3 | 46.2 | 14.5 | 11.6 | 55.9 | 7.4 |
| DiffGesture | 18.6 | 37.1 | 12.6 | 13.8 | 38.9 | 7.0 |
| Ours | **7.3** | **52.7** | **4.9** | **6.5** | **67.4** | **4.2** |

Table 6: Results on the AIST++ Dance and BHS Dance datasets with **generation quality** metrics.

## 5.3 Analysis and Ablation Study

**User Study.** We conducted a user study with 20 annotators on the AIST++ Dance dataset to evaluate the generation performance. For each method, 200 samples were generated, and 20 paired samples were randomly selected for each comparison group. Annotators were asked to respond on site: "*Which dance/music is more realistic and matches the music/dance better?*". The human evaluation results, shown in Figure 6, indicate that our method outperforms SOTA approaches in both motion-to-music and music-to-motion generations. Notably, a preference drop is observed when BiCoR-VAE is not employed, highlighting the importance of an aligned latent space for cross-modal generation.

**Motion Encoding.** For motion sequence encoding, we compare the spatial position-based method with the directional vector-based method, which learns the unit directional vectors of the given adjacency set, and reconstructs the human pose with the calculated mean bone lengths [48]. However, as shown in Table 7 (#1), the spatial position-based method proved superior, likely due to the error introduced by movements that alter bone length, such as squatting and bending.

| Id | Method | Music Metrics | | Motion Metrics | |
|---|---|---|---|---|---|
| | | FAD ↓ | F1 ↑ | FID ↓ | F1 ↑ |
| #1 | Ours w/ Directional Vectors | 10.9 | 91.4 | 14.7 | 38.0 |
| #2 | Ours w/o Mel-spectrogram | 12.8 | 95.6 | 9.5 | 41.6 |
| #3 | Ours w/o Rhythmic Contrastive Learning (RCL) | 8.5 | 93.1 | 8.1 | 37.9 |
| #4 | Ours w/o Feed-Forward Transformer (FFT) | 11.0 | 95.8 | 11.6 | 41.4 |
| #5 | Ours (Joint Generation) | 8.1 | 96.5 | 8.8 | 45.4 |
| #6 | Ours (Joint Generation&Variable Length) | 9.1 | 97.6 | 8.5 | 49.6 |
| #7 | Ours (Cross Generation) | 8.9 | 98.1 | 7.3 | 46.2 |

Table 7: Ablation study on motion-to-music and music-to-motion generations. We use the FAD/FID as the quality assessment and the F1 score as the beat-matching assessment.

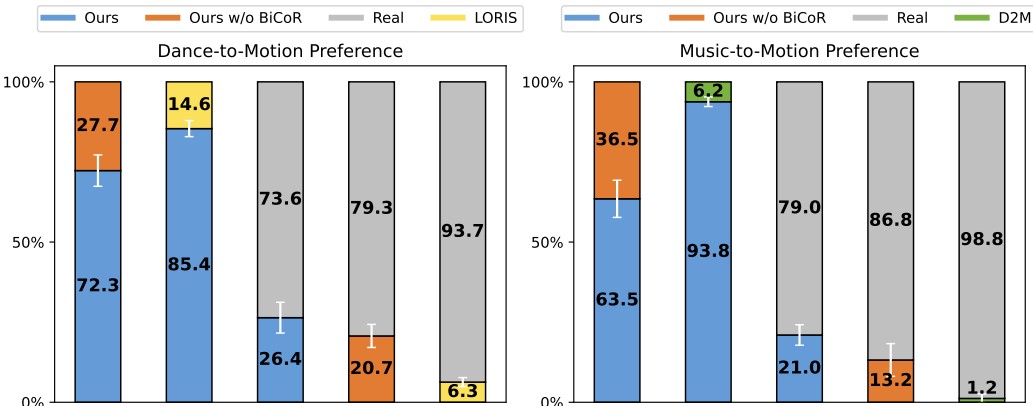

Figure 6: Results of human evaluation on motion-to-music and music-to-motion generations.

**Music Encoding.** For music encoding, we evaluated the spectrogram-based method against the raw waveform-based method. According to Table 7 (#2), the raw waveform-based method gains performance declines in both FAD and F1 metrics. This is attributed to the lengthy audio sequences introduced by the raw waveform, introducing difficulties for diffusion modeling training.

**Learning Techniques.** In MoMu-Diffusion, there are two key learning techniques: rhythmic contrastive learning (RCL) and Feed-Forward Transformer (FFT). From Table 7, we can observe that "Ours w/o RCL" gains a clear drop on the beat-matching metric F1 (#3) and "Ours w/o FFT" gains a drop on the synthesis quality metric FID/FAD (#4), respectively. "Ours w/o FFT" means we use a U-Net backbone for the diffusion model, which has been shown inferior to our FFT-based model in long sequence modeling. Equipped with both RCL and FFT, MoMu-Diffusion ensures both generation quality and cross-modal alignment.

**Joint Generation in Variable Length.** MoMu-Diffusion supports multi-modal joint generation in variable lengths, facilitated by a "pad-and-truncate" strategy in the diffusion model and the proposed cross-guidance sampling. To validate this capability, 1000 samples with varying lengths (10-30 seconds) are generated using different Gaussian noise vectors. With a cross-guidance sampling timestep set to $T_c = 0.5T$, Table 7 (#5, #6), we can find that for multi-modal joint generation, MoMu-Diffusion shows that MoMu-Diffusion achieves comparable performance to the conditional models with clean condition inputs and advanced performance on the joint generation scenario. More ablation studies are provided in Appendix D.

## 6 Conclusion

In this paper, we propose MoMu-Diffusion, the first multi-modal framework designed to learn the long-term synchronization and correspondence between human motions and music. In MoMu-Diffusion, we have two key designs: bidirectional contrastive rhythmic VAE (BiCoR-VAE) for learning modality-aligned latent spaces and Transformer-based diffusion model for learning long-term dependencies. Through extensive experiments, we demonstrate MoMu-Diffusion's efficacy across motion-to-music, music-to-motion, and joint motion-music generations.

## Acknowledgments

This work was supported by National Natural Science Foundation of China under Grant No. 62222211. This work was also supported by National Natural Science Foundation of China under Grant No.62072397.

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

# Appendices

## MoMu-Diffusion: On Learning Long-Term Motion-Music Synchronization and Correspondence

## A  Implementation Details of BiCoR-VAE

### A.1  Model Configurations

For BiCoR-VAE, we use an encoder-decoder architecture, in which the 1D convolutional network and spatial transformer are used. The input is downsampled by a Conv1d downsampling layer and then forwarded to the middle block, finally upsampled by a Conv1d upsampling layer. The detailed hyper-parameters of BiCoR-VAE are listed in Table 8.

### A.2  Model Training

We use a two-stage training strategy for BiCoR-VAE. Firstly, we train the mel-spectrogram VAE with three loss functions: reconstruction loss $\mathcal{L}_{recon}$, KL loss $\mathcal{L}_{KL}$ and a GAN loss $\mathcal{L}_{GAN}$ to prevent over-smoothed mel-spectrogram:

$$\mathcal{L}_{stage1} = \mathcal{L}_{recon} + \lambda_1 \mathcal{L}_{KL} + \lambda_2 \mathcal{L}_{GAN}, \tag{10}$$

where $\lambda_1$ is set to 1e-5 and $\lambda_2$ is set to 0.5. Note that the GAN loss contains two steps: it first updates the generator part (mel-spectrogram VAE) with $\mathcal{L}_{stage1}$, and then updates the additional discriminator. After training mel-spectrogram VAE, we freeze it, and train the motion VAE with the proposed contrastive rhythmic learning loss defined in Eq (4) in stage 2:

$$\mathcal{L}_{stage2} = \mathcal{L}_{recon} + \lambda_3 \mathcal{L}_{KL} + \lambda_4 \mathcal{L}_{contrast}, \tag{11}$$

where $\lambda_3$ is set to 1e-5 and $\lambda_4$ is set to 1. For training BiCoR-VAE, we use the AdamW optimizer with a learning rate of 2e-4 and training epochs of 300. We use 8 NVIDIA 4090 GPUs and it takes about 12 hours to finish. To decode the mel-spectrogram into high-fidelity music, we use the BigvGAN model [28] pretrained on the AudioSet dataset [14].

## B  Implementation Details of Cross-Modal Generation

### B.1  Dataset

For motion-to-music, we evaluate our method on the latest LORIS benchmark [49], which contains 86.43 hours of video samples with paired music. This benchmark incorporates three challenging scenarios: dancing, floor exercise, and figure skating. For dancing, 1,881 25-second videos are collected from AIST++ [30], a fine-annotated subset of the dancing dataset AIST [44]. For floor exercise, 1,950 25-second and 660 50-second videos are collected from the Ginegym dataset [42]. For figure skating, 8,585 25-second and 4,147 50-second videos are collected from the FisV [47] and FS1000 [46] datasets.

For music-to-motion, we use two datasets: AIST++ Dance [30] and BHS Dance. About 71 hours BHS Dance videos are collected from [26], which contains three dancing types: "Ballet", "Zumba", and "Hip-Hop". In our experiments, each dataset is randomly split with a 90%/5%/5% proportion for training, validation, and testing.

Note that only the AIST++ Dance dataset is used for both motion-to-music and music-to-motion generations. This is because the Floor Exercise and Figure Skating datasets involved too heavy motion variation, which makes it hard for the pose extraction algorithm to extract the high-accuracy motion sequences. As for the BHS Dance dataset, it only provides the MFCC audio features without raw audio. Therefore, we can not conduct motion-to-music experiments on it.

### B.2  Data Processing

We use mel-spectrogram as audio feature representation, We first resample the audio to 16kHz. We use 80 filters with fft set to 1024 and hop length set to 256 while processing the mel spectrogram

| Hyper-Parameters | BiCoR-VAE |
|---|---|
| Hidden channels | 20 |
| Residual blocks | 2 |
| Channel multiplier | [1,2,4] |
| Spatial attention layers | 3 |
| Downsampling rate | 2 |
| Kernel size of Conv1d | 5 |
| Total Params | 213M |

Table 8: Hyper-parameters of the BiCoR-VAE model.

| Hyper-Parameters | MoMu-Diffusion |
|---|---|
| Dimension of conditional embedding | 1024 |
| Input channels | 20 |
| Dimension of Hidden representation | 576 |
| Number of attention heads | 8 |
| Number of Transformer blocks | 4 |
| Kernel size of Conv1d projection network | 5 |
| Padding of Conv1d projection network | 3 |
| Diffusion Steps | 1000 |
| Total Params | 158M |

Table 9: Hyper-parameters of the FFT model.

using Hann window with a window size of 1024. For human motions, OpenPose [3] is applied to extract 2D body keypoints, and can process a video at 60 fps. We use the pre-trained Body-25 model to extract 25 key points of the human body, but some key points are difficult to extract consistently and some are less relevant to actions. As implemented by [26], we finally choose the 14 most relevant keypoints to represent the poses, i.e., nose, neck, left and right shoulders, elbows, wrists, hips, knees, and ankles. We interpolate the missing detected keypoints from the neighboring frames so that there are no missing keypoints in all extracted clips.

## B.3 Model Configurations

For the denoising part, we use the Transformer backbone rather than the U-Net. The hyper-parameters of our FFT model are listed in Table 9. The FFT diffusion model is trained by the AdamW optimizer [23] with a learning rate of 1.6e-5 and a lambda linear scheduler with a warmup step of 10000. We train the diffusion model with 200 epochs for each task. It takes about 2 days for 8 NVIDIA 4090 GPUs. For the Figure Skating dataset, it takes about 4 days since this dataset is large.

## B.4 Evaluation Metrics: Motion-to-Music

To evaluate whether the synthesized music is aligned with the given motion, we use the improved **Beats Coverage Scores (BCS)** and **Beat Hit Scores (BHS)** to validate the rhythm correspondence and cross-modal alignment of synthesized music. The improved BCS and BHS are first proposed by [4, 26], then used for rhythmic dance-to-music validation [53, 52], and improved by [49] for long-term rhythmic music validation. Also, we report **Coverage Standard Deviation(CSD)** and **Hit Standard Deviation(CSD)** to evaluate the robustness of generative models. Finally, the **F1** scores of improved BCS and BHS are also reported as an overall assessment. BCS and BHS are designed by computing matching degrees of the rhythm points from synthesized music and ground-truth music. Let $N_s$ be the rhythm point number of synthesized music, $N_t$ be the rhythm point number of ground-truth music, and $N_m$ be the number of matched rhythm points, the BCS is defined as $BCS = N_m/N_s$ and the BHS is defined as $BHS = N_m/N_t$, respectively. However, these metrics are not suitable for long-term music evaluations since 1) the second-wise rhythm detection algorithm leads to an extremely sparse vector and 2) BHS can easily exceed 1 if the rhythm points of generated music are more than ground truth. Therefore we use an improved audio onset detection

**Algorithm 1:** Pseudo code for cross-modal (motion-to-music) sampling.

---

**Input:** The latent mel-spectrogram representation $z_a$, latent motion representation $z_m$, the
pre-trained denoiser $\theta_a$ for motion-to-music, and the decoder $D_a$ for mel-spectrogram.

$t \leftarrow T$,
$z_a(t) \leftarrow \mathcal{N}(0, \mathbf{I})$
**while** $t > 0$ **do**
   |   $z_a(t) \leftarrow$ sample from $p_{\theta_a}(z_a(t - \Delta t)|z_a(t), t, z_m)$
   |   $t \leftarrow t - \Delta t$
**end**
**return** $D_a(\hat{z_a})$

---

algorithm to avoid sparse rhythm vectors. Here is the Python code based on the Librosa library:
*librosa.onset.onset_detect(y=audio, sr=sampling_rate, wait=1, delta=0.2, pre_avg=3, post_avg=3, pre_max=3, post_max=3, units='time').*

For validating the quality of synthesized music, we use the Fréchet Audio Distance (**FAD**) and the **Diveristy** score. We use the pre-trained VGGish model from `https://github.com/gudgud96/frechet-audio-distance` to compute the FAD scores. Based on the feature extractor VGGish, we compute the Diversity score by using the average feature distance for paired samples. Specifically, the Diversity score contains inter-diversity and intra-diversity. Inter-diversity is obtained by computing the average feature distance between 200 combinations of 50 pieces of music from different motions and the intra-diversity is obtained by computing the average feature distance between all combinations of 5 pieces of music from the same motion input.

### B.5 Evaluation Metrics: Music-to-Motion

To evaluate whether the synthesized motion is aligned with the reference music, we also use these five beat-matching metrics. However, since the number of musical beats is always more than the number of kinematic beats in real-world products, we use the musical beats as the reference for evaluation, which is also consistent with previous works [26]. Concretely, Let $N_s$ be the kinematic point number of synthesized motion, $N_t$ be the rhythm point number of ground-truth music, and $N_m$ be the number of matched points, the BCS is defined as $BCS = N_m/N_s$ and the BHS is defined as $BHS = N_m/N_t$, respectively. For kinematic beat extraction, we use the bin-wise directrogram difference (defined in Eq (2)) as the indicator [4].

For validating the quality of synthesized motion, we use the Fréchet Inception Distance (**FID**) and the **Diveristy** score. To compute the FID score, we follow the design of [26] and train a motion classifier on the BHS Dance dataset with three classification categories: "Ballet", "Zumba", and "Hip-Hop". The motion classifier consists of a MotionBert encoder [51] and a classification head. The motion classifier is trained by an Adam optimizer with a learning rate of 1e-4 and 100 epochs. Then, we use the trained MotionBert encoder as the feature extractor for computing the FID and Diversity scores. The definition of Diversity score here is the same as Appendix B.4.

## C Pseudo Codes

We provide the pseudo-codes of cross-modal generation and multi-modal joint generation in Algorithm 1 and 2, respectively. For the cross-modal generation, we take the motion-to-music as an example while the implementations of music-to-motion are symmetrical.

## D The Choice of $T_c$ in Joint Generation

We propose a simple cross-guidance sampling strategy to combine multiple cross-modal generative models for joint generation. In this process, there is a hyper-parameter $T_c$ that controls the modality fusion timestep. In Table 10, we study five variants: $T_c$ begins from $0.9T$ to $0.1T$ with an interval of $0.2T$. From these results, we can observe that employing the cross-guidance strategy in the early sampling steps is not feasible since the predicted latent representation contains too many noises. However, we can find that MoMu-Diffusion ($T_c = 0.7T$) obtains an acceptable performance,

**Algorithm 2:** Pseudo code for multi-modal joint sampling.

---

**Input:** The latent mel-spectrogram representation $z_a$, latent motion representation $z_m$, the pre-trained denoisers $\theta_a$ and $\theta_m$, the decoders $D_a$ and $D_m$, the cross-guidance scale $\gamma$, the pre-defined schedule $\alpha_1, ..., \alpha_T$ and $\overline{\alpha}_t = \prod_{i=1}^{t} \alpha_i$.

$t \leftarrow T$,
$T_c \leftarrow \gamma T$
$z_a(t) \leftarrow \mathcal{N}(0, \mathbf{I})$
$z_m(t) \leftarrow \mathcal{N}(0, \mathbf{I})$
**while** $T \geq t > T_c$ **do**
  $\quad z_a(t - \Delta t) \leftarrow$ sample from $p_{\theta_a}(z_a(t - \Delta t)|z_a(t), t, \emptyset)$
  $\quad z_m(t - \Delta t) \leftarrow$ sample from $p_{\theta_m}(z_m(t - \Delta t)|z_m(t), t, \emptyset)$
  $\quad t \leftarrow t - \Delta t$
**end**
**while** $T_c \geq t > 0$ **do**
  $\quad \hat{z_a} = \frac{z_a(t)}{\sqrt{\overline{\alpha}_t}} - \frac{\sqrt{1-\overline{\alpha}_t}}{\sqrt{\overline{\alpha}_t}} \epsilon_{\theta_a}(z_a(t), t, \emptyset)$
  $\quad \hat{z_m} = \frac{z_m(t)}{\sqrt{\overline{\alpha}_t}} - \frac{\sqrt{1-\overline{\alpha}_t}}{\sqrt{\overline{\alpha}_t}} \epsilon_{\theta_m}(z_m(t), t, \emptyset)$
  $\quad z_a(t - \Delta t) \leftarrow$ sample from $p_{\theta_a}(z_a(t - \Delta t)|z_a(t), t, \hat{z_m})$
  $\quad z_m(t - \Delta t) \leftarrow$ sample from $p_{\theta_m}(z_m(t - \Delta t)|z_m(t), t, \hat{z_a})$
  $\quad t \leftarrow t - \Delta t$
**end**
**return** $D_a(\hat{z_a})$, $D_m(\hat{z_m})$

---

| Method | Joint Generation | Music Metrics | | Motion Metrics | |
|---|---|---|---|---|---|
| | | FAD ↓ | F1 ↑ | FID ↓ | F1 ↑ |
| MoMu-Diffusion ($T_c = 0.9T$) | ✓ | 14.8 | 82.4 | 17.3 | 25.8 |
| MoMu-Diffusion ($T_c = 0.7T$) | ✓ | 10.9 | 95.9 | 9.7 | 37.8 |
| MoMu-Diffusion ($T_c = 0.5T$) | ✓ | 8.1 | 96.5 | 8.8 | 45.4 |
| MoMu-Diffusion ($T_c = 0.3T$) | ✓ | 7.5 | 90.4 | 9.0 | 42.1 |
| MoMu-Diffusion ($T_c = 0.1T$) | ✓ | 8.0 | 85.5 | 9.5 | 32.6 |

Table 10: Ablation study of the cross-guidance step $T_c$ on the AIST++ Dance dataset.

indicating that the denoising process is coarse-to-fine. Using fewer cross-guidance sampling steps (like $T_c = 0.1T$) can ensure the quality of generated samples, but the cross-modal alignment is omitted, leading to a low F1 score. Therefore, we use $T_c = 0.5T$ in our paper to trade off the sampling quality and cross-modal alignment in joint generation.

# E   Failure Cases

Since our method predicts pose points, the deviation between points can cause abnormal length of human skeleton. Three sets of examples are shown in Figure 7, with anomalous frames on the left and the corrected frames on the right.

We first calculated the average bone length between each pair of keypoints in the dataset, after which we post-processed the predicted keypoints. We specify a threshold (which we specify as 1.3 times the mean bone length) and correct the predicted bone lengths to the mean bone length once they exceed the threshold, an approach that significantly enhances model generation. It is worth noticing that the post-processing can not fully address this issue but alleviate it.

# F   More Qualitative Results

We provide more qualitative results in Figure 8, 9, 10, and 11

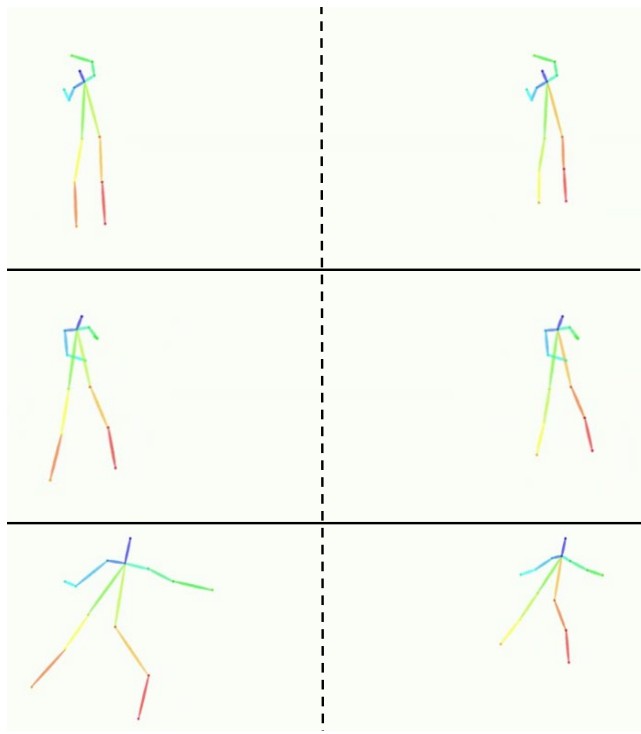

Figure 7: Three failure cases and the corrected results.

## G  Used Resources and Licenses

In this paper, we use several open resources, including `https://github.com/OpenGVLab/LORIS`(All Rights Reserved), `https://github.com/NVlabs/Dancing2Music` (NVIDIA Source Code License, 1-Way Commercial), `https://github.com/gudgud96/frechet-audio-distance` (All Rights Reserved), `https://github.com/Walter0807/MotionBERT` (All Rights Reserved), and `https://github.com/Text-to-Audio/Make-An-Audio` (All Rights Reserved). We use these resources for research purpose only.

## H  Limitations and Boarder Impact

Due to the limited motion-music data and computing resources, the scaling law of our model is not testified in a super large dataset. Besides, our model depends on some data pre-processing methods like mel-spectrogram extraction and keypoints extraction by OpenPose, which may lead to error accumulations.

MoMu-Diifusion promotes both neural motion and music synthesis, so it may help expand any impact that generative systems have on the broader world like copyright conflicts. We will add constraints and licenses when open-resourcing our code and pre-trained models.

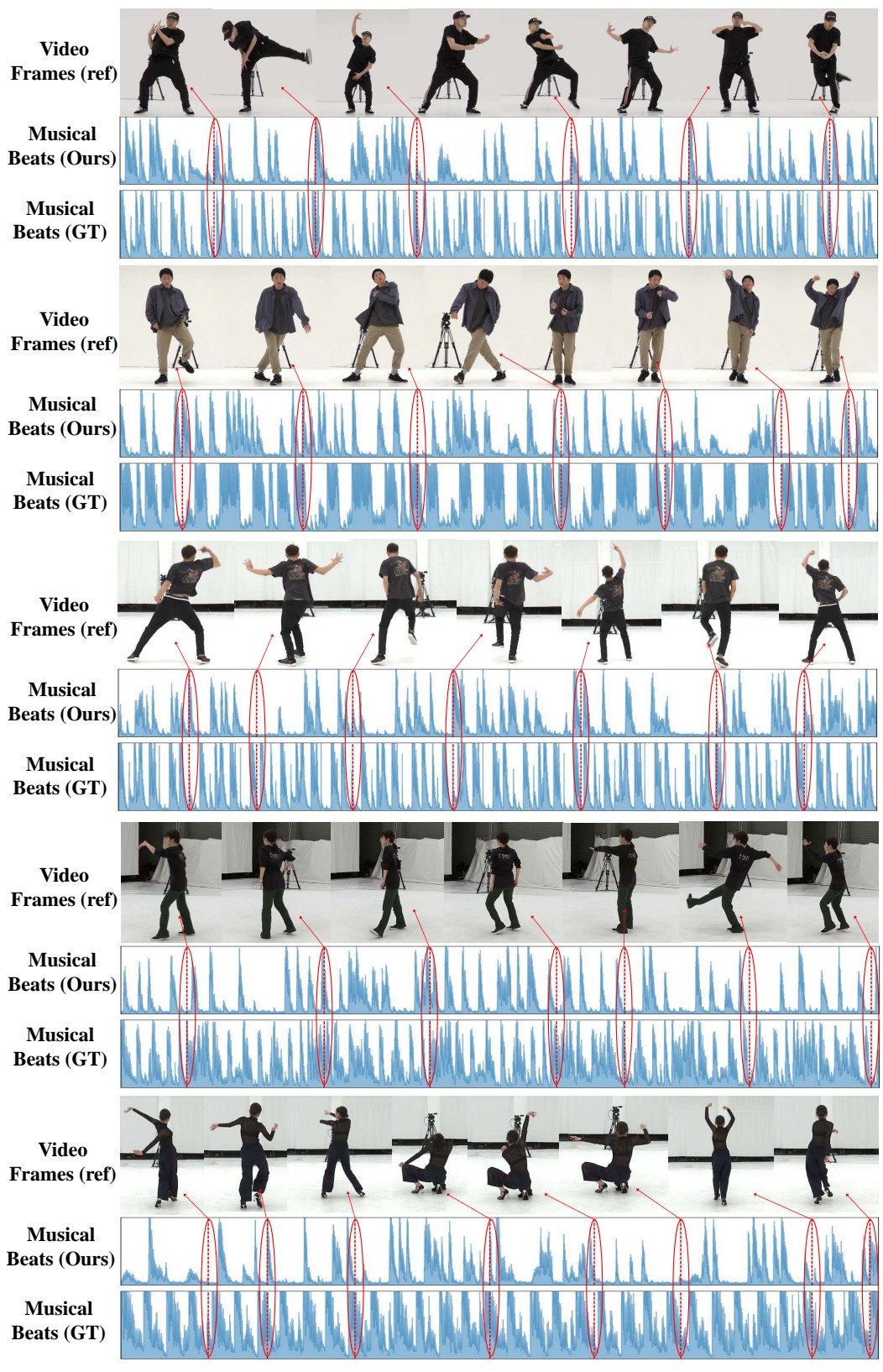

Figure 8: Example of beat matching on the AIST++ Dance (motion-to-music). The red dashes indicate the extracted musical beats. The red arrow points to the video frame at that particular moment.

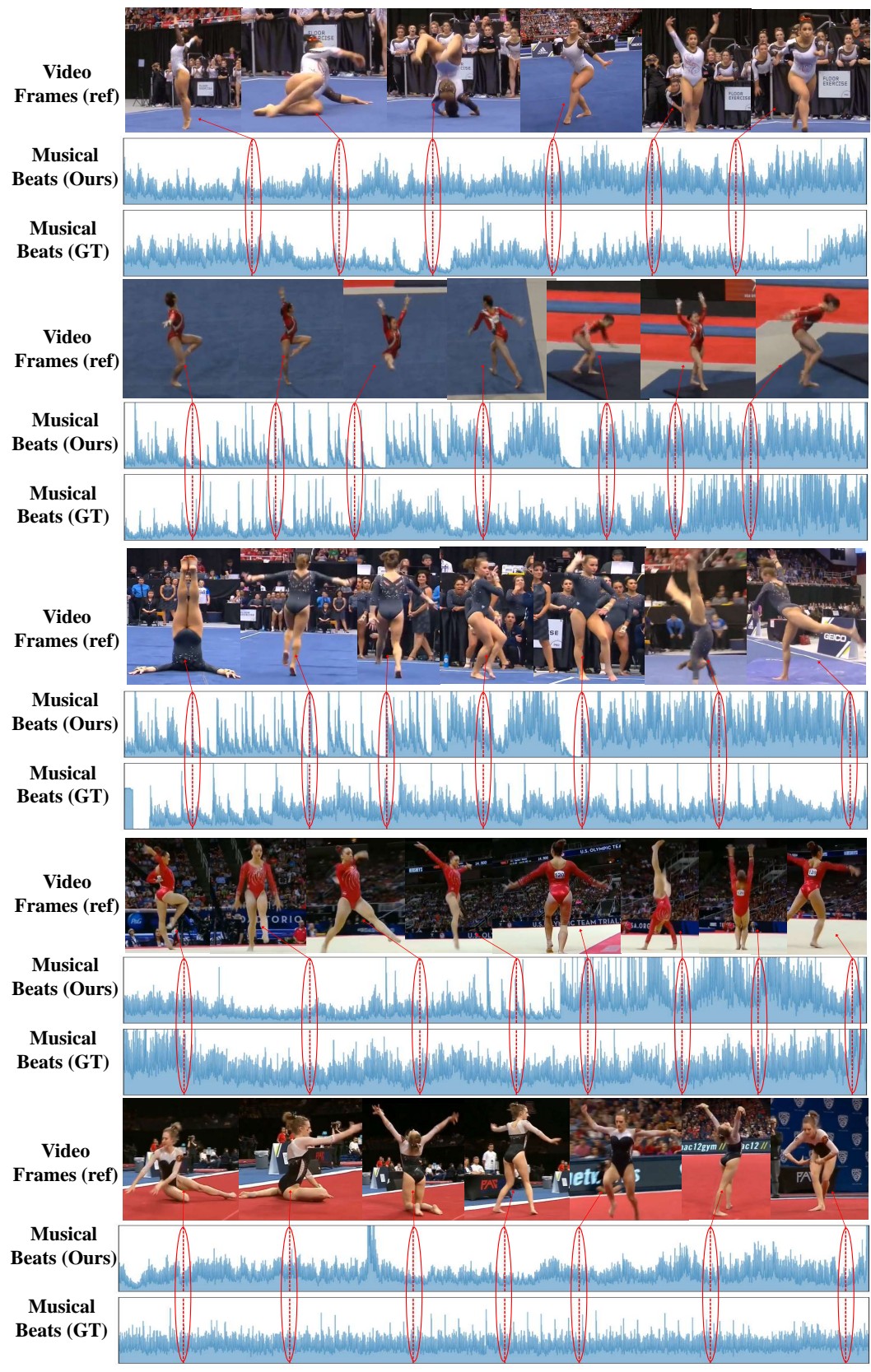

Figure 9: Example of beat matching on the Floor Exercise (motion-to-music). The red dashes indicate the extracted musical beats. The red arrow points to the video frame at that particular moment.

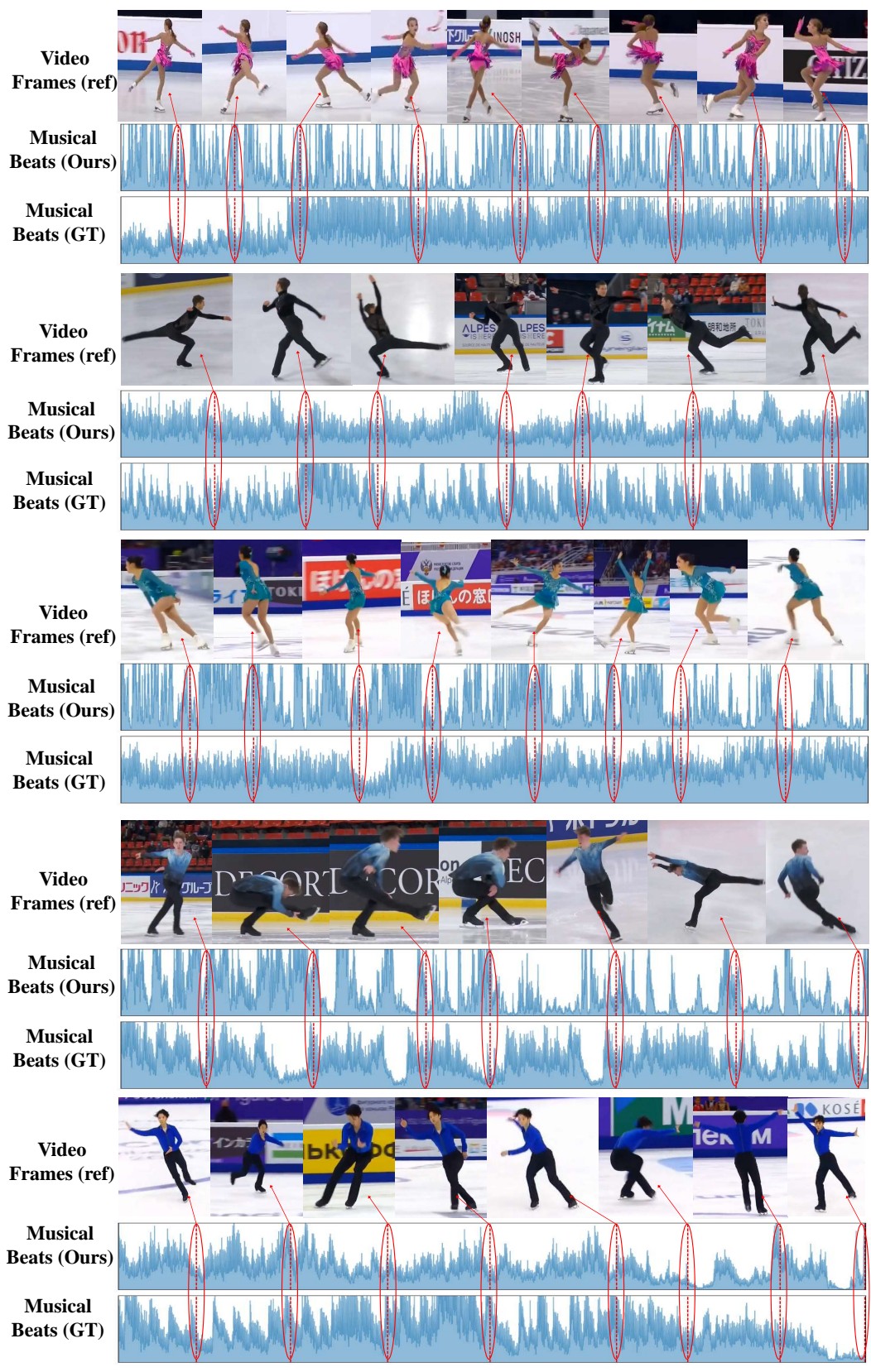

Figure 10: Example of beat matching on the Figure Skating (motion-to-music). The red dashes indicate the extracted musical beats. The red arrow points to the video frame at that particular moment.

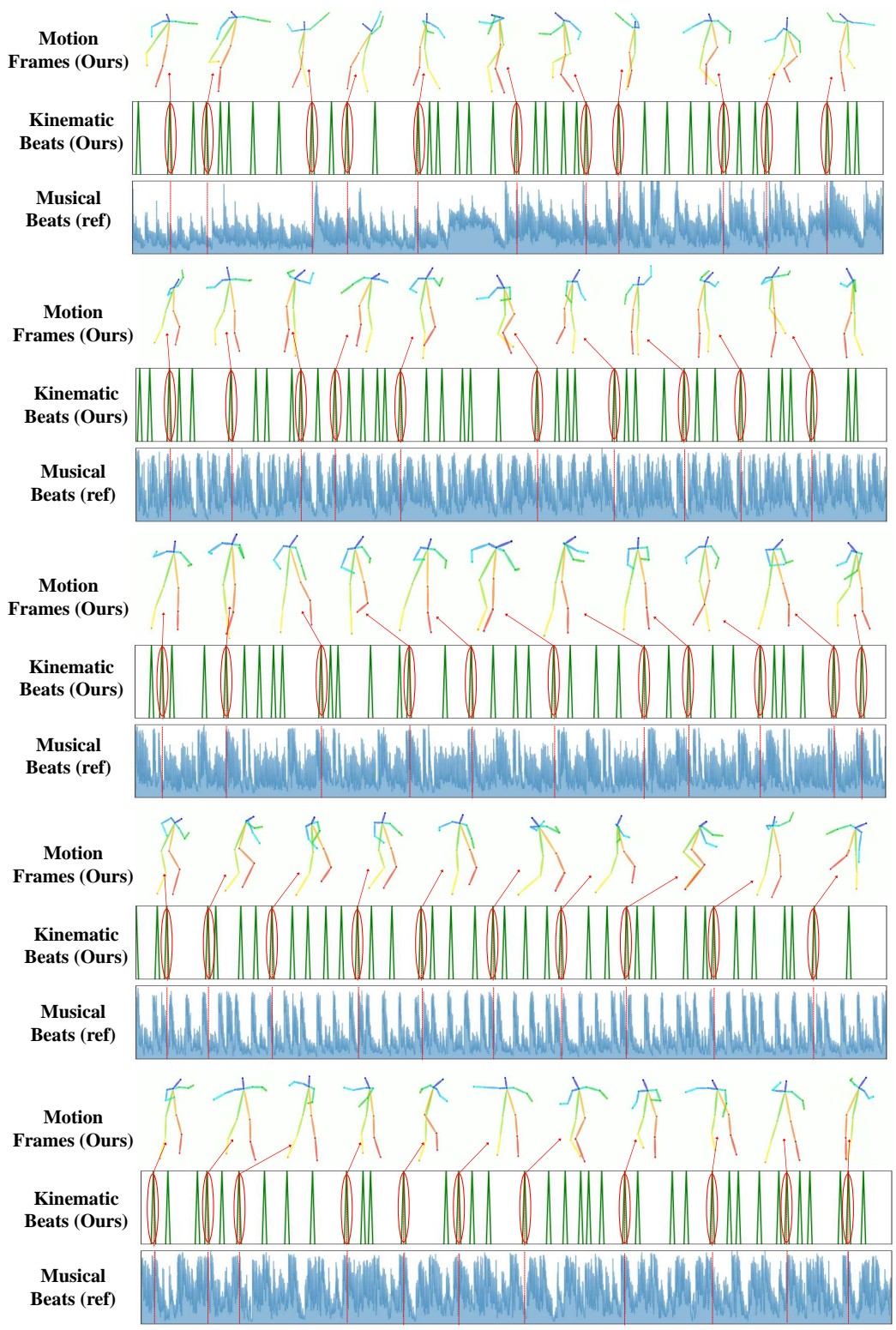

Figure 11: Example of beat matching on the AIST++ Dance (music-to-motion). The red dashes indicate the extracted kinematic beats of the synthesized motion. The red arrow points to the frame of the synthesized motion sequence at that particular moment.

