# OpenReview forum: "MoMu-Diffusion: On Learning Long-Term Motion-Music Synchronization and Correspondence"
_NeurIPS.cc/2024/Conference — NeurIPS 2024 poster_

### Official Review · Reviewer_aP7b · 2024-07-01

**Soundness:** 3
**Presentation:** 3
**Contribution:** 3
**Rating:** 6
**Confidence:** 2

**Summary:**

The authors proposed a novel framework to address the motion-to-music and music-to-motion tasks. They leveraged aligned latent spaces between motion and music, a multi-modal diffusion transformer, and a cross-guidance sampling strategy. Experiments were conducted to demonstrate that their approach outperforms their counterpart models.

**Strengths:**

The paper proposes several interesting concepts, including rhythmic contrastive learning to produce aligned latent spaces across different modalities and transformer-based diffusion models to address music-to-motion and motion-to-music synthesis tasks. Additionally, it offers the possibility to jointly synthesize music and motion. The user study and ablation study are also well conducted.

**Weaknesses:**

Please refer to the questions section.

**Questions:**

- The authors discussed the architectures of the multi-modal VAE in Section 3.1. Could the authors provide more specific implementation details, including
  - Whether the motion vectors in shape $\mathbb{R}^{T_m\times J\times 2}$ are first of all flattened to $\mathbb{R}^{T_m\times(J\times 2)}$ for VAE? If not, how are they processed?
  - What exactly are $T_a$ and $T_u$ in Line 105 for the music VAE? Does this imply that the music VAE has a lower downsampling ratio than the motion VAE in the temporal dimension?
  - Is normalization needed for the autoencoder representations of the both modalities so that they can be further processed by the diffusion model?
- The authors went through how they constructed the motion-music latent clip for VAE training which incorporates the kinematic amplitude indicator as depicted in Eq. 2. Could the authors further elaborate on why incorporating this is important by e.g., showing a histogram of kinematic amplitude in the dataset and how it is sampled?

**Limitations:**

The authors have adequately address the limitations.

---

> ### Author Rebuttal · Authors · 2024-08-05
>
> We thank the reviewer for his/her insightful comments and for appreciating the technical contributions of our paper. Here, we respond to your questions as follows:
>
> ***1. More specific implementation details***
>
> a) The motion vectors are flattened to $n \in \mathbb{R}^{T_m\times(J\times2)}$ first.
>
> b)  No. We use the same downsampling rate for both motion and music inputs (Table 8 in the Appendix) and align the frame rates by pre-processing techniques.
>
> c) Yes. The latent representations from different modalities are normalized before forwarding the diffusion model.
>
> Thanks for your comments. We will add these explanations in the revised manuscript to make it clearer.
>
> ***2. Why incorporate the kinematic amplitude indicator and how the latent clip is sampled?***
>
> Why: In lines 111-117, we mentioned that existing audio-visual temporal contrastive pre-training methods are not feasible for motion-music alignment. For example, the temporal contrast, which seeks to maximize the similarity of audio-visual pairs from the same time segment while minimizing the similarity of pairs from different segments, may push away two clips that sampled from different time segments but shared similar rhythmic beats. That is why we incorporate the kinematic amplitude indicator to enable motion-music contrastive pre-training. As for the illustration of kinematic amplitude, you can refer to the line chart example in Figure 2(a), whose values are normalized within the range of (0,1).
>
> How: In Figure 2(a), we have vividly illustrated the sampling process. For a motion-music latent representation pair, we compute the 2D motion directogram according to Eq. (1) first, and then derive the kinematic amplitude indicator for each temporal step by aggregating the bin-wise directogram difference according to Eq. (2). Finally, we randomly sample the motion-music clips from different categories to construct the contrastive learning objective according to Eq. (4). Note that the classification of the different categories is based on the kinematic amplitude indicator values, which undergo in-clip max-pooling, as described by Eq. (3). We will include the example in Figure 2 along with its accompanying explanations to make the Section 3.2 easier to understand.

---

> > ### Comment · Reviewer_aP7b · 2024-08-12
> >
> > Thank you for your insightful comments. I will maintain my score.

---

### Official Review · Reviewer_1QHY · 2024-07-11

**Soundness:** 3
**Presentation:** 3
**Contribution:** 3
**Rating:** 6
**Confidence:** 4

**Summary:**

MoMu-Diffusion is a motion-music co-generation model with an aim of improved temporal synchronization of the generated motion and music sequence, based on two major components:
1. a bidirectional contrastive rhythmic VAE (BiCoR-VAE) that provides aligned latent space of the motion and music through joint training, and
2. a multi-modal diffusion transformer (DiT) the model the latent space with cross-guidance sampling.

**Strengths:**

The proposed BiCoR-VAE's rhythmic contrastive objective using a kinematic amplitude indicator is well inspired and interesting approach to extract key frames that contain motions of interest. It is based on the valid assumption that the motions in the video data likely align with the beat present in the music.

The beat matching metrics in Table 2 justifies the improvements made in the proposed approach, and the ablation studies has been done thoroughly in Table 7. Human evaluation in Figure 6 shows strong prererence to the proposed model.

**Weaknesses:**

I would like to see the qualitative samples of the baselines and the ablation models to the demo page as well to form the readers opinion on the claimed synchronization of this work. While the presented results in the manuscript looks convincing, having direct comparisons in the demo will be helpful for the readers to have a better idea of the improvements.

**Questions:**

* (line 121-122): I think it would be helpful to add a specific example of the framerate differece between motion and music frames. Currently it might read like the motion framerate is always higher (by integer multiple) than the mel spectrogram framerate.
* Assuming that the framerate design is defined by the mel framerate (followed by the vocoder), is resampling the motion sequence to match the mel framerate trivial enough to assume that we have the time aligned motion and music frames in the pre-processing stage?

**Limitations:**

The authors have mentioned the limitations in this work, including limited computational and data budget to verify the idea to scaled up dataset that potentially include video clips that do not adhere to the assumptions made in the rhythmic contrastive objective.

---

> ### Author Rebuttal · Authors · 2024-08-05
>
> We thank the reviewer for his/her insightful comments and for appreciating the technical contributions of our paper. Here, we respond to your questions as follows:
>
> ***1. Comparison with other methods on the demo page.***
>
> Thanks for your suggestions. Since the illustrated video samples are pretty long, the existing demo page has exceeded 567 MB. We will work to add the generated samples of SoTA methods: LORIS in motion-to-music and D2M in music-to-motion and re-organize the demo page. Currently, you can refer to their released samples from https://justinyuu.github.io/LORIS and   https://github.com/NVlabs/Dancing2Music for a preliminary qualitative comparison.
>
> ***2. Add a specific example of the different frame rates of motion and music inputs.***
>
> Assuming that we have a ten-second video and process it at 60 fps for pose sequence extraction, which results in 600 motion frames. For the corresponding audio, we resample it to 16khz and extract the mel-spectrogram with a hop length of $256$. Then, the audio frames are $10\times16000/256=625$. Finally, to align the frames from different modalities, we evenly drop $25$ audio frames to ensure the temporal dimensions of motion and music inputs are equal. In the revised manuscript, we will add an example to make it clearer.
>
> ***3. Frame rate alignment between motion and music inputs in the pre-processing stage.***
>
> Due to the different sampling rates of video and audio, it is hard to ensure the audio and visual inputs are strictly temporally aligned. We ensure that the timing error is within a small range when we implement cross-modal temporal alignment in the pre-processing stage. For example, following the example mentioned above, we denote the audio frame rate as $f_1$ and the motion sequence frame rate as $f_2<f_1$. The minimum timing error is $|1/f_1-1/f_2|$ and the maximum timing error is $|k/f_1-k/f_2|$, where $k$ is the dropping interval and is computed by $k/f_1=(k-1)/f_2$. Recalling that $f_1=16000/256$ and $f_2=60$, the range of timing error is ($0.6$ms, $17$ms). For human beings, a timing error of less than $30$ms is typically hard to capture. Also, we will improve the pre-processing techniques to reduce the timing error in future work.
>
> Thanks again for your valuable comments. We are happy to engage in discussion if you have any further questions.

---

> > ### Comment · Reviewer_1QHY · 2024-08-12
> >
> > Thank you for your rebuttal and clarifications from my questions. Documenting the specific example regarding the design of framerate would provide clearer understanding of this work.
> >
> > I am retaining my score since my overall positive assessment has not changed. Thanks again for your rebuttal.

---

### Official Review · Reviewer_pw59 · 2024-07-14

**Soundness:** 3
**Presentation:** 3
**Contribution:** 3
**Rating:** 7
**Confidence:** 4

**Summary:**

The paper propose MoMu-Diffusion, that generates both music-to-motion and motion-to-music videos. MoMu-Diffusion achieves SOTA results and is able to generate joint distribution of music and motion instead of one way.

**Strengths:**

1. Generate joint distribution of music and motion
2. Propose rhythmic contrastive learning to align music and motion.
3. Demo included, reasonable evaluation metrics and great experiments.

**Weaknesses:**

1. I might miss it, but that would be nice if there is an ablation study of the hidden size and layers of Diffusion Transformer.

**Questions:**

1. How long can you generate the video in the joint distribution generation setting? Is that possible to use DiffCollage (Zhang et al., 2023) for longer video generation?

**Limitations:**

1. Maybe can explore longer video generations.

---

> ### Author Rebuttal · Authors · 2024-08-05
>
> We thank the reviewer for his/her insightful comments and for appreciating the technical contributions of our paper. Here, we respond to your questions as follows:
>
> ***1. Ablation study of the hidden size and layers of Transformer***
>
> We appreciate your suggestion on the ablation study of model size. Due to the time limit, we may be not able to present comprehensive results (e.g., model params $\textgreater$ $1$B) in rebuttal within a short response period. We implement on the AIST++ dataset and study four model configurations: the combination of hidden size in $[576,768]$ and Transformer layer in $[4,8]$:
>
> |Hidden size | Layers | Params | Beat-F1 (mo2mu) $\uparrow$| FAD (mo2mu) $\downarrow$| Beat-F1 (mu2mo)$\uparrow$ | FID (mu2mo) $\downarrow$|
> |-----------|---------|---------|---------|---------|---------|---------|
> | 576 | 4 | 158M | 98.1 | 8.9 | 46.2 | 7.3|
> |576 | 8 | 315M | 98.3 | 9.0 | 46.7 | 6.8 |
> | 768 | 4 | 281M | 98.1 | 8.7 | 47.5 | 7.5 |
> 768 | 8 | 559M | 98.7 | 7.2 | 52.1 | 6.2 |
>
>
> We can observe that there is a slow performance boost when scaling the model size. The scaling law usually suggests that the performance can be improved by scaling up the training data and model size, simultaneously. It is reasonable to hypothesize that the bottleneck for the performance of motion-music generation lies in the data size. For existing public datasets like AIST++, LORIS, and BHS Dance, we think the base model configuration (DiT: 158M, BiCoR-VAE: 213M) is matched.
>
> ***2. How long can we generate the video in the joint generation setting?***
>
> In joint generation, the length of the generated video is suggested to be less than 25 seconds, which is mainly caused by the video length of training data (i.e., AIST++). Currently, for motion-music generation, the major difficulty in achieving longer video generation is the lack of corresponding training data. In terms of algorithm design, It is reasonable to speculate that DiffCollage is effective in dealing with longer video generation (more than 1min) thanks to its parallel sampling and smaller sequence length. For the experiments in our paper, the end-to-end diffusion formulation can effectively address the problems and it is not necessary to split the sequence into several clips. In future work, we are going to build a larger dataset with longer video length and are willing to incorporate DiffCollage for a faster and longer generation.
>
> Thanks again for your valuable comments. We are happy to engage in discussion if you have any further questions.

---

> > ### Comment · Reviewer_pw59 · 2024-08-07
> >
> > Thank you for your experiments and insightful explanations. I maintain my score.

---

### Official Review · Reviewer_E4Pj · 2024-07-15

**Soundness:** 4
**Presentation:** 3
**Contribution:** 4
**Rating:** 9
**Confidence:** 4

**Summary:**

This paper proposes a framework that enables the generation of music from motion, motion from music, or both simultaneously while maintaining synchrony between the two. To achieve this, kinematic amplitude is extracted from the motion, and motion audio clip segments corresponding to different kinetic amplitudes are sampled. Contrastive loss is applied to the audio and motion embeddings of these sampled segments to ensure that the embeddings align with each kinetic amplitude. The resulting embeddings are then used for joint diffusion, enabling audio-to-motion, motion-to-audio, or joint generation. A method for applying cross guidance is also proposed. Quantitative evaluations were conducted based on the beat information of the generated music, as well as the FID and diversity of the generated motion. A user study demonstrated that the generated outputs were preferred over those from previous models.

**Strengths:**

The paper proposes an embedding method using kinetic amplitude and contrastive learning to ensure that the intensity of motion and music/motion are well-aligned. This approach effectively leverages prior knowledge of motion. I believe this methodology can be applied to problems requiring synchronization of intensity and timing beyond just motion-music synchronization. The use of diffusion for multimodal generation, incorporating DIT and introducing a cross-guidance sampling strategy, also holds potential for future research in multimodal generation.

**Weaknesses:**

The proposed methodology leverages the relationship between changes in motion and beats in music as the primary element of synchronization, with evaluations mainly focused on this aspect. However, there can be instances where the beatness in motion or music is not clear (especially during certain periods). A discussion on how to handle such cases, or considerations beyond beatness, would be beneficial for achieving more natural generation.

**Questions:**

Suggestion: Section 3.2 is a crucial part that explains BiCoR-VAE, but the process of sampling clips was difficult to understand. Referring to Figure 2(a) made it easier. Including an example to explain this process would make it more comprehensible.

**Limitations:**

-

---

> ### Author Rebuttal · Authors · 2024-08-05
>
> We thank the reviewer for his/her insightful comments and for appreciating the technical contributions of our paper. Here, we respond to your questions as follows:
>
> ***1. How to handle the cases where the beat in motion or music is not clear and achieve/evaluate natural generation beyond beat alignment?***
>
> Training Phase: The diffusion model's goal is to learn the conditional distribution between motion and music, encompassing not only beat alignment but also the synchronization of temporal and semantic elements. While additional alignments beyond the beat are not explicitly modeled during the pre-training phase, they are implicitly achieved through the diffusion model. The incorporation of supplementary learning objectives, such as semantic contrast as utilized in Diff-Foley, into the BiCoR-VAE could lead to challenges in hyper-parameter tuning and optimization. Therefore, we only emphasize the beat alignment in the pre-training stage and implicitly model the cases where the beat is not clear in the diffusion training. In future work, we will conduct an in-depth study on stably combining both beat and non-beat alignments during the pre-training stage.
>
> Evaluation Phase: In evaluating the quality of the generated samples, we have implemented a suite of beat-matching metrics (e.g., beat F1 score) and metrics that assess the overall generation quality (e.g., FAD). Compared to introducing additional objective metrics, we posit that the design of subjective evaluations holds greater significance in assessing motion-music generation. Drawing a parallel with the well-researched domain of text-to-speech synthesis, despite numerous objective metrics such as word error rate and speaker similarity, subjective assessments are still significant for measuring the naturalness of synthesized samples. Consequently, in the context of motion-music generation, we have conducted subjective A/B testing to compare our approach with existing state-of-the-art methodologies, as depicted in Figure 6.
>
> ***2. Including an example to make the Section 3.2 easier to understand.***
>
> Thanks for your suggestions. For a motion-music latent representation pair, we compute the 2D motion directogram according to Eq. (1) first, and then derive the kinematic amplitude indicator for each temporal step by aggregating the bin-wise directogram difference according to Eq. (2). Finally, we randomly sample the motion-music clips from different categories to construct the contrastive learning objective according to Eq. (4). Note that the classification of the different categories is based on the kinematic amplitude indicator values, which undergo in-clip max-pooling, as described by Eq. (3). We will include the example in Figure 2 along with its accompanying explanations to make the Section 3.2 easier to understand.
>
>
> Thanks again for your valuable comments. We are happy to engage in discussion if you have any further questions.

---

### Decision · Program_Chairs · 2024-09-25

**Decision:**

Accept (poster)

**Comment:**

All reviewers find the paper interesting and novel. They unanimously agree that the paper warrants an acceptance. There are a few minor but still valuable suggestions, and I highly recommend the authors polish the paper further to make it more complete.